# Biomarkers of Neutrophil Activation in Patients with Symptomatic Chronic Peripheral Artery Disease Predict Worse Cardiovascular Outcome

**DOI:** 10.3390/biomedicines11030866

**Published:** 2023-03-12

**Authors:** Giacomo Buso, Elisabetta Faggin, Alessandro Bressan, Silvia Galliazzo, Francesco Cinetto, Carla Felice, Michele Fusaro, Andreas Erdmann, Paolo Pauletto, Marcello Rattazzi, Lucia Mazzolai

**Affiliations:** 1Angiology Division, Heart and Vessel Department, Lausanne University Hospital, University of Lausanne, 1011 Lausanne, Switzerland; 2Department of Clinical and Experimental Sciences, 2a Medicina—Azienda Socio Sanitaria Territoriale Spedali Civili di Brescia, University of Brescia, 25100 Brescia, Italy; 3Department of Medicine—DIMED, University of Padua, 35122 Padua, Italy; 4Department of Internal Medicine Unit, Ospedale S. Valentino, 31044 Montebelluna, Italy; 5Medicina Interna I, Ca’ Foncello University Hospital, 31100 Treviso, Italy; 6Department of Radiology, Ca’ Foncello University Hospital, 31100 Treviso, Italy; 7Ospedale Riabilitativo di Alta Specializzazione (ORAS)-ULSS 2 TV, Motta di Livenza, 31045 Treviso, Italy

**Keywords:** peripheral artery disease, biomarkers, neutrophil degranulation, myeloperoxidase, neutrophil extracellular traps

## Abstract

Neutrophils play a role in cardiovascular (CV) disease. However, relatively scant evidence exists in the setting of peripheral artery disease (PAD). The aims of this study were to measure biomarkers of neutrophil activation in patients with symptomatic chronic PAD compared with healthy controls, to assess their association with PAD severity, and to evaluate their prognostic value in patients with PAD. The following circulating markers of neutrophil degranulation were tested: polymorphonuclear neutrophil (PMN) elastase, neutrophil gelatinase-associated lipocalin (NGAL), and myeloperoxidase (MPO). Neutrophil extracellular traps (NETs) were quantified by measuring circulating MPO–DNA complexes. Patients with PAD underwent a comprehensive series of vascular tests. The occurrence of 6-month major adverse CV (MACE) and limb events (MALE) was assessed. Overall, 110 participants were included, 66 of which had PAD. After adjustment for conventional CV risk factors, PMN-elastase (adjusted odds ratio [OR]: 1.008; 95% confidence interval [CI]: 1.002–1.015; *p* = 0.006), NGAL (adjusted OR: 1.045; 95%CI: 1.024–1.066; *p* < 0.001), and MPO (adjusted OR: 1.013; 95%CI: 1.001–1.024; *p* = 0.028) were significantly associated with PAD presence. PMN-elastase (adjusted hazard ratio [HR]: 1.010; 95%CI: 1.000–1.020; *p* = 0.040) and MPO (adjusted HR: 1.027; 95%CI: 1.004–1.051; *p* = 0.019) were predictive of 6-month MACE and/or MALE. MPO displayed fair prognostic performance on receiver operating characteristic (ROC) curve analyses, with an area under the curve (AUC) of 0.74 (95%CI: 0.56–0.91) and a sensitivity and specificity of 0.80 and 0.65, respectively, for a cut-off of 108.37 ng/mL. MPO–DNA showed a weak inverse correlation with transcutaneous oximetry (TcPO2) on proximal foot (adjusted ρ −0.287; *p* = 0.032). In conclusion, in patients with symptomatic chronic PAD, enhanced neutrophil activity may be associated with an increased risk of acute CV events, rather than correlate with disease severity. Further research is needed to clarify the role of neutrophils in PAD natural history.

## 1. Introduction

Peripheral artery disease (PAD) is a prevalent condition that affects over 200 million people around the world, including 40 million people in European countries [1]. This condition, which is most often secondary to atherosclerosis, is characterized by reduced blood flow and oxygen supply to the lower limbs due to progressive narrowing of the arteries. This can impair muscle function and overall quality of life, as well as significantly increase the risk of limb loss in affected patients.

From a pathophysiological perspective, circulating leukocytes, particularly neutrophils, have been shown to play a crucial role in atherosclerosis and atherothrombosis [2,3]. Neutrophils are the most abundant type of leukocytes and represent the first line of defense of innate immunity, capable of capturing and destroying microorganisms through phagocytosis and intracellular degradation. They also participate as mediators of inflammation [4].

Accumulation of neutrophils in atherosclerotic plaques increases plaque rupture risk. In particular, activated neutrophils may release proteolytic enzymes through degranulation, promoting endothelial cell detachment and exposure of sub-endothelial collagen and fibronectin to platelets, thus leading to plaque destabilization [5].

Moreover, recent studies showed that neutrophils stimulated by microbes, inflammatory agents, reactive oxygen species or activated platelets, release nuclear material, forming a web-like extracellular network known as neutrophil extracellular traps (NETs), made of DNA, histones, and granule constituents. In vitro and animal studies suggest that NETs may play a role in thrombus organization/stability [6,7,8], endothelial damage [9], and atherosclerosis progression [10]. Some human studies also demonstrated that markers of NETs, such as nucleosomes and myeloperoxidase (MPO)–DNA complexes, are increased in patients with severe coronary atherosclerosis and predict risk of cardiac events [11]. Similar findings were observed in patients with venous thromboembolism (VTE) [12,13,14].

Although the role of neutrophil activation, including neutrophil degranulation and NETs release, in cardiovascular (CV) disease is well-documented, relatively scarce information exists in the context of symptomatic chronic PAD to date [15].

Accordingly, the aims of the present study were to measure markers of neutrophil activation in patients with symptomatic chronic PAD compared with healthy controls, evaluate their association with disease severity, and assess their prognostic value.

## 2. Materials and Methods

### 2.1. Study Design and Participants

This is a multicenter, multinational, prospective case-control study conducted in the University Hospital of Lausanne (“Centre Hospitalier Universitaire Vaudois”, CHUV), Switzerland, in collaboration with both the University Hospital of Padua and the University Hospital of Treviso in Italy.

In the Angiology Division of the CHUV, consecutive patients aged 40–75 years with an established diagnosis of symptomatic chronic PAD were recruited. Symptomatic chronic PAD was defined as a Leriche–Fontaine stage from II to IV [16], associated with an ankle-brachial index (ABI) ≤ 0.9, or a toe-brachial index (TBI) ≤ 0.7 (in case of incompressible arteries).

Sex and BMI-matched subjects without evidence of thrombotic events and coronary artery disease (CAD) were enrolled as a control group at Cà Foncello University Hospital of Treviso in Italy. In particular, we recruited patients without a history of CV disease that underwent computed tomography angiography scan by using a “triple rule out protocol” for atypical chest pain to exclude the presence of pulmonary thromboembolism, CAD, and acute aortic dissection. As for controls, the inclusion criteria were: electrocardiogram free of abnormalities compatible with ischemia, necrosis, or myocardial injury; negative troponin enzyme levels; baseline left ventricular ejection fraction ≥ 45%; absence of history of cardiopathy and cardiac arrhythmias including atrial fibrillation, supraventricular tachycardia, and extrasystoles; estimated mortality risk < 2% according to the GRACE Risk Score.

Overall exclusion criteria were: active or previous diagnosis of cancer; presence of rheumatic disease or immunosuppressive therapy (including chronic steroid treatment); concomitant exacerbations of chronic disease (such as chronic obstructive pulmonary disease, inflammatory bowel disease or immunological disorders); clinical and biochemical evidence of active infectious disease; recent surgery (less than three months); pregnancy.

The aims of the present study were: (1) to measure levels of circulating markers of neutrophil activation (neutrophil degranulation and neutrophil-derived NETs) in patients with symptomatic chronic PAD compared with healthy controls; (2) to explore the association between circulating markers of neutrophil activation and PAD severity; and (3) to evaluate the predictive value of circulating markers of neutrophil activation for adverse outcomes at 6-month follow-up in patients with symptomatic chronic PAD.

Research has been performed in accordance with the ethical guidelines of the 1975 Declaration of Helsinki and local ethic committee-approved study protocol (CER-VD, BASEC Project-ID: 2016-01250). All subjects gave their written, informed consent upon enrollment in the study.

### 2.2. Clinical Data Collection and Biochemical Analysis at Baseline

The following data were collected from each participant: age; sex; height; weight; body mass index (BMI, calculated as body weight [kg] divided by the square of the height [m]); family history of atherosclerotic CV events (ischemic stroke, transient ischemic attack, acute coronary syndrome); presence of diabetes (defined as a documented diagnosis of diabetes or patient on antidiabetic treatment with no other clear indication), and self-reported history of active smoking (if more than 100 cigarettes lifetime); current use of antihypertensive drugs, antiplatelet drugs, and lipid lowering treatment (statins).

In patients with symptomatic chronic PAD, Leriche–Fontaine stage, arterial sites involved (aortoiliac, femoral-popliteal, and below-knee arteries), as well as personal history of CV disease (CAD and cerebrovascular disease [CeVD]), VTE, and lower limb revascularization, were recorded. At enrollment, the blood pressure (BP) (mmHg) was assessed in all participants as the mean of three measures one minute apart. Hypertension was defined as systolic BP (SBP) ≥ 140 and/or diastolic BP (DBP) ≥ 90 mmHg or ongoing antihypertensive medications. All the subjects underwent fasting blood sampling to measure creatinine level (mg/dl) and glomerular filtration rate (GFR) (ml/min/1.73 m^2^), estimated through the 2009 Chronic Kidney Disease Epidemiology Collaboration (CKD-EPI) creatinine equation. Total cholesterol, low-density lipoprotein (LDL) and high-density lipoprotein (HDL) cholesterol, as well as triglycerides (mg/dl), were measured. Blood serum samples were collected from each patient and stored at −70 °C until further analysis.

### 2.3. Quantification of Circulating Markers of Neutrophil Activation and NETs Target by ELISA

Dedicated enzyme-linked immunosorbent assay (ELISA) kits were used for quantitative measurement of neutrophil degranulation markers, including polymorphonuclear neutrophil (PMN)-elastase (ab119553; Abcam, Cambrige, UK), neutrophil gelatinase-associated lipocalin (NGAL) (KIT 036RUO; BioPorto Diagnostics A/S, Hellerup, Denmark), and MPO (BMS2038INST; Invitrogen, Waltham, MA, US).

NETs-associated MPO–DNA complexes were measured by adding serum into 96-well plates coated with a monoclonal anti-human MPO antibody (MABX4043-10KC; Millipore, Burlington, MA, US) after saturation of no specific binding site with bovine serum albumin, followed by incubation with peroxidase-labelled anti-DNA monoclonal antibody included in the Cell Death ELISA kit (11774425001; Roche Diagnostics, Mannheim, Germany). The optical absorbance was measured at 405 wavelength by using Berthold Mithras multimode reader (Berthold Technologies GmbH, Bad Wildbad, Germany).

### 2.4. Vascular Assessment in Patients with PAD

All study participants with symptomatic chronic PAD underwent a comprehensive series of vascular tests at enrollment (V0) and 6 months later (V1). The visit V0 included: ABI calculation (and TBI calculation, in case of incompressible arteries); transcutaneous oximetry (TcPO2) measurement on the distal and proximal foot; constant-load treadmill test; 6-min walking test; pulse wave velocity (PWV); and flow-mediated dilatation (FMD) of the brachial artery. The visit V1 included ABI calculation (and TBI calculation, in case of incompressible arteries) and 6-min walking test. Details of such tests are provided in Supplementary Methods.

### 2.5. Clinical Outcomes

At the 6-month follow-up, the following outcomes were assessed in patients with PAD: (1) presence of major adverse CV (MACE) and/or limb events (MALE); (2) ABI reduction ≥ 0.15 or (or TBI reduction ≥ 0.1, in case of incompressible arteries); and (3) 6-min maximal walking distance (6MWD) reduction ≥ 20 m.

MACE was defined as the composite of all-cause death, non-fatal myocardial infarction, non-fatal stroke, and heart failure leading to hospitalization, as well as coronary revascularization, including percutaneous coronary intervention, and coronary artery bypass graft. Deaths were regarded to be attributable to a cardiac cause unless a non-cardiac death could be confirmed. MALE was defined as disabling claudication or severe limb ischemia leading to an intervention as well as major vascular amputation. Only patients with no intercurrent lower limb revascularization were included in the analysis of the remaining two outcomes, as surgery could have improved both ABI (or TBI) and 6MWD.

All patients who did not show up for the V1 were interviewed by telephone in order to ascertain their clinical conditions as well as the development of MACE and MALE. If patients did not attend the scheduled follow-up visit planned and were not reachable by telephone, they were considered as lost to follow-up.

### 2.6. Statistical Analysis

Patients were classified according to whether they had PAD or not. First, we compared groups in terms of demographics, concomitant diseases, and laboratory data. According to the normality of the distribution, the Student’s t-test or the Mann–Whitney test were used to compare groups in terms of continuous variables, whereas the Chi-square test or the Kruskal-Wallis test were applied for categorical variables. Continuous variables were reported as mean ± standard deviation (SD), regardless of the normality of the distribution, for the sake of simplicity. Categorical variables were reported as a percentage.

We then performed univariable logistic regression for associations between clinical variables and PAD status, expressed as odds ratios (ORs) with a 95% confidence interval (CI). As for multivariable regression, covariates entering the models were selected by a significance level of *p* < 0.10 on bivariate analysis using the stepwise regression model with backward elimination.

In patients with PAD, we further performed linear correlation analysis with the estimation of the Pearson (or Spearman, for those variables found not to follow a normal distribution) coefficient (ρ) to test the association between the levels of circulating markers of neutrophil degranulation and NETs and the above vascular parameters assessed on V0. Multiple linear regression analysis was carried out including biomarkers selected by a significance level of *p* < 0.10 on bivariate analysis adjusted for covariates with a well-known correlation reported in the literature.

We then performed a multivariable analysis using a Cox model to identify the predictors for MACE and/or MALE during the first 6 months of follow-up. As for the other outcomes (assessed at a fixed time point of 6 months from enrollment), univariable logistic regression was carried out to identify their predictors at baseline. Covariates entering into the models were selected by a significance level of *p* < 0.10 on bivariate analysis or by a well-known association reported in the literature.

For all biomarkers showing significant predictive ability in terms of MACE and/or MALE at multivariable analysis, the prognostic performance and optimal value for identifying patients at risk were defined based on receiver operating characteristic (ROC) curves by calculating the area under the curve (AUC), sensitivity, specificity, and Youden’s index (= sensitivity + specificity − 1).

All the statistical tests were two-tailed and conducted at a significance level of 0.05.

Sample size has been estimated on the basis of previous investigation performed in patients with VTE [12] and CAD [11].

We conducted statistical analyses using SPSS (IBM SPSS Statistics for Windows, v. 25.0. IBM Corp., Armonk, NY, USA).

## 3. Results

### 3.1. General Study Population Characteristics

From 13 October 2016 to 18 May 2020, 110 participants were included in the study, of which 66 had symptomatic chronic PAD and 44 were healthy controls. General participants’ characteristics are summarized in Table 1. The mean age was 59 (±11) years and the mean BMI was 27.3 (±5) kg/m^2^. Twenty-four (21.8%) participants had a BMI compatible with obesity class I or more (according to the World Health Organization classification). Overall, no significant difference was observed between participants with and without PAD in terms of BMI, waist, and obesity rates. Male sex was prevalent in both groups (68.2 vs. 72.2%, respectively; *p* = 0.766), while patients with PAD were significantly older (64 vs. 51 years; *p* < 0.001) and were more likely to have diabetes (36.4 vs. 4.7%; *p* < 0.001), hypertension (69.7 vs. 40.9%; *p* = 0.003), and active smoking (51.5 vs. 29.5%; *p* = 0.037).

The Leriche–Fontaine stage was IIa in 50 (75.8%) and IIb in 15 (22.7%) patients, whereas one patient displayed PAD stage IV. Twenty-six (39.4%) patients with PAD had aortoiliac involvement, while 51 (77.3%) and 20 (30.3%) of them had femoral-popliteal and below-knee arteries involvement, respectively. Thirty-three (50%) patients with PAD had a history of previous lower limb revascularization. At baseline, mean ABI and TBI were 0.72 (±0.13) and 0.54 (±0.14), respectively. Mean TcPO2 was 41 (±12) and 37 (±12) mmHg on distal and proximal foot, respectively.

As for the laboratory parameters at baseline, total and LDL cholesterol levels were significantly lower in patients with PAD than healthy controls (*p* < 0.001 each), while no significant differences were found between groups in terms of creatinine and GFR, HDL cholesterol, and triglycerides (Table 1).

### 3.2. Circulating Markers of Neutrophil Degranulation in Patients with PAD and Controls

Levels of all markers of neutrophil degranulation were significantly higher in patients with symptomatic chronic PAD, compared with healthy controls (*p* < 0.001 for each) (Table 1, Figure 1).

Uni- and multivariable regression analyses, the latter including age, diabetes, hypertension, and active smoking as covariates, confirmed a significant association with the presence of PAD for PMN-elastase (adjusted OR: 1.008; 95%CI: 1.002–1.015; *p* = 0.006), NGAL (adjusted OR: 1.045; 95%CI: 1.024–1.066; *p* < 0.001), and MPO (adjusted OR: 1.013; 95%CI: 1.001–1.024; *p* = 0.028).

Correlation analyses between the above markers and PAD severity showed a significant, though weak, negative correlation between NGAL and the following parameters: 6-min pain-free walking distance (6PFWD) (ρ −0.298; *p =* 0.025), pain-free walking distance (PFWD) (ρ −0.256; *p* = 0.046), and maximal walking distance (MWD) (ρ −0.305; *p* = 0.017). No significant correlation was found between NGAL and 6MWD (ρ −0.243; *p* = 0.066), as well as between the other markers of neutrophil activation and further vascular tests (Figure 2, Appendix A).

After inclusion in a multivariable regression analysis model including age, diabetes, and active smoking, no significant correlation was found between NGAL and the above parameters of PAD severity.

### 3.3. Circulating NETs in Patients with PAD and Controls

Levels of MPO–DNA were not significantly different in patients with symptomatic chronic PAD compared with healthy controls (*p* = 0.241) (Table 1, Figure 1), which was also confirmed by univariable regression analysis (OR: 1.022; 95%CI: 0.969–1.077; *p* = 0.426).

Correlation analyses in patients with PAD found a significant, though weak, negative correlation between MPO–DNA and TcPO2 on proximal foot (ρ −0.284; *p* = 0.029) (Figure 2, Appendix A), which remained significant after inclusion in a multivariable regression analysis model including age, diabetes, and active smoking (adjusted ρ −0.287; *p* = 0.032).

### 3.4. Clinical Outcomes in Patients with PAD

Of the included 66 patients with symptomatic chronic PAD, three were lost to follow-up. Within the first 6 months of follow-up, 11 of the remaining 63 patients developed MACE and/or MALE (two myocardial infarctions, and nine disabling claudication or severe limb ischemia leading to an intervention). After exclusion of patients with symptomatic chronic PAD who underwent intercurrent limb revascularization or were lost to follow-up, data on ABI at V1 were available on 49 patients; 14 of them had a significant ABI (≥0.15) or TBI (≥0.1) reduction at 6 months. As for the 6MWT, data on 6MWD at V1 were available on 40 patients, 10 of whom had a 6MWD reduction ≥ 20 m at 6 months.

Compared to patients without MACE and/or MALE, those who experienced this outcome displayed significantly higher levels of both MPO (150.17 vs 98.52 ng/mL, respectively; *p* = 0.020) and MPO–DNA (0.362 vs 0.256 abs 405–490; *p* = 0.014) at baseline (Table 2), whereas clinical features including age, PAD staging, comorbidities, and baseline treatment were similar between groups (Appendix A).

Multivariable Cox regression analyses found that both PMN-elastase (adjusted hazard ratio [HR]: 1.010; 95%CI: 1.000–1.020; *p* = 0.040) and MPO (adjusted HR: 1.027; 95%CI: 1.004–1.051; *p* = 0.019) were predictive of MACE and/or MALE at 6-month follow-up. Conversely, no significant association was found between the above markers at baseline and both ABI reduction ≥ 0.15 (or TBI reduction ≥ 0.1, in patients with incompressible arteries) and 6MWD reduction ≥ 20 m at 6-month follow-up (Table 3).

In terms of MACE and/or MALE, ROC curve analyses showed fair and poor prognostic performance for MPO and PMN-elastase, respectively. In particular, MPO displayed an AUC of 0.74 (95%CI: 0.56–0.91), as well as a sensitivity and specificity of 0.80 and 0.65, respectively, for a cut-off of 108.37 ng/mL, determined using Youden’s index. Lower values of AUC were found for PMN-elastase, as shown in Figure 3.

## 4. Discussion

Patients with symptomatic chronic PAD showed significantly increased levels of circulating markers of neutrophil degranulation compared to healthy subjects, whereas levels of MPO–DNA, a specific marker of NETs, were similar in the two groups. In patients with symptomatic chronic PAD, markers of neutrophil degranulation (PMN-elastase and MPO) were predictive of worse CV outcome at 6 months. Conversely, none of the neutrophil activation markers correlated strongly with a series of PAD severity parameters at baseline, nor were they significantly associated with a reduction of ABI or 6MWD at follow-up.

MPO is a heme-containing peroxidase stored in neutrophilic granules and released upon neutrophil activation. This extracellular protease is expressed in atherosclerotic lesions [17], and MPO-containing macrophages were found to be particularly abundant in vulnerable and ruptured plaques [18]. In humans, MPO deficiency was found to be associated with a reduced risk of CV disease [19]. Conversely, individuals with genetic polymorphisms of MPO are at an increased risk of CAD [20]. Circulating levels of MPO are associated with both the presence [21] and severity [22,23] of CAD on angiography. Moreover, both in patients presenting to an emergency department with chest pain [24] and those with established CAD diagnosis [23], MPO circulating levels are predictive of adverse outcomes. Of note, inhibition of MPO improved blood flow in a diabetic mouse model of hind-limb ischemia [25], while higher serum MPO levels were associated with increased risk of myocardial infarction or stroke in patients with PAD [26]. Intriguingly enough, in a recent prospective cohort study including patients with PAD undergoing digital subtraction angiography [27], baseline MPO levels were 3.68 times higher in patients with all-cause death and MACE, and 1.48 times higher in those with MALE than those without such outcomes at 24-months follow-up. Moreover, the authors found higher MPO levels in patients with multi-bed vascular disease compared to those with PAD alone, suggesting that this biomarker may reflect the extent of vascular damage in patients with atherosclerotic disease [27].

In line with these findings, our study found higher levels of MPO and other circulating markers of neutrophil degranulation in patients with symptomatic chronic PAD than in heathy subjects. Adjustment for conventional risk factors associated with PAD presence on bivariate analysis, namely age, diabetes, hypertension, and active smoking, did not attenuate the association, indicating that these markers are not strongly associated with such covariates. Importantly, two of the markers of neutrophil degranulation, namely MPO and PMN-elastase, were found to be predictive of MACE and/or MALE within the following 6 months. In particular, ROC curve analyses showed fair prognostic performance for the former, underlining their possible role in CV risk definition in patients with PAD.

Neutrophil elastase plays a crucial role in bacterial killing [28] and seems to be involved in several noninfectious diseases, such as respiratory diseases and arthritis [29], whereas its role in CV disease is less clear. Higher levels of circulating neutrophil elastase have been reported in patients with myocardial infarction [30], while an animal model suggested that this protease enhances myocardial injury by inducing an excessive inflammatory response in cardiomyocytes, thus worsening the prognosis post-myocardial infarction [31]. Our study seems to suggest a role for PMN-elastase in the natural history of PAD as well, although further research will be needed to clarify this aspect.

In addition to classical strategies such as degranulation, neutrophils are also able to release their nuclear contents into the extracellular space. This chromatin mesh called NETs is highly cytotoxic [27], and recent evidence revealed a potential role of NETs in linking sterile inflammation with thrombosis, including atherothrombosis [32,33]. In a murine model of myocardial ischemia/reperfusion, NET-induced no-reflow was described [34], whereas another study showed significantly reduced infarct size in protein arginine deiminase 4 (PAD4) knockout mice with myocardial infarction, which are unable to undergo at least one form of NETs generation [35]. In humans, NETs are an abundant component of coronary thrombi [36], and coronary NETs burden is a strong independent predictor of adverse clinical outcome in patients with ST-elevation myocardial infarction [36]. Moreover, NETs excess may be associated with impaired wound healing and seems to predict poor wound outcomes in patients with diabetes [37]. In the setting of PAD, NETs have been associated with several ischemic outcomes, particularly in patients undergoing peripheral angioplasty and stenting [38]. In human research, several surrogate markers of NETs generation have been used so far, including citrullinated histones, complexes of double-stranded DNA (dsDNA), and MPO–DNA complexes. Nonetheless, the specificity of dsDNA in this setting may be questionable, since extracellular nucleosomes may be released during necrosis and apoptosis of cells other than neutrophils [39], whereas the role of histone citrullination in NET generation is still under debate [40]. Accordingly, MPO–DNA should be a more reliable, specific marker of NETs generation. In this respect, previous studies found that MPO–DNA is a component of coronary thrombus in acute myocardial infarction [41] and correlates with the presence of severe coronary atherosclerosis [11]. Moreover, MPO–DNA may be associated with both CV disease severity and prognosis [42,43].

In our study, MPO–DNA levels were not significantly higher in patients with symptomatic chronic PAD compared with healthy subjects. As most of the above studies on NETs were carried out in the setting of acute vascular events, NETs generation may therefore not be significantly enhanced in patients with chronic, stable CV disease, including PAD. Another hypothesis to be considered in this respect concerns the use of statins. Besides decreasing cholesterol synthesis, these drugs are known for a series of pleiotropic effects, including antioxidant and inflammatory properties, improvement of endothelial function, and stabilization of atherosclerotic plaques [44]. Importantly, increasing evidence has demonstrated that statins have modulatory effects on neutrophil function, and particularly on NETs generation [45,46]. Furthermore, research on patients with carotid artery stenosis demonstrated lower circulating NGAL levels in those on statins compared to the non-statin group [47], whereas no difference was found between patient with and without statin treatment in terms of serum MPO and neutrophil elastase [48]. Such evidence could play a relevant role in the interpretation of our results. In particular, the widespread use of statins in patients with PAD might have influenced the circulating levels of MPO–DNA complexes and significantly attenuated differences between groups. Since our patient cohort is too small to speculate further on this, future studies on larger cohorts with a greater representation of patients not receiving statin therapy may clarify these issues.

Of note, MPO–DNA was not predictive of MACE and/or MALE at Cox regression analysis in our study, though its levels at baseline were significantly higher in patients developing such an outcome within the next 6 months. Further research on larger cohorts using longer follow-up periods is warranted to clarify this apparent contradiction.

In order to evaluate lower limb perfusion, walking functional capacity, arterial stiffness, and endothelial function, we performed a comprehensive assessment of the patients’ vascular status at baseline. Previous research reported that MPO is strongly associated with both ABI values and PAD presence [49]. However, no strong correlation was found between markers of neutrophil activation, including MPO and these vascular tests. At multivariable regression analysis, only MPO–DNA was significantly, though weakly, inversely correlated with TcPO2 on proximal foot, whereas no significant correlation was found with TcPO2 on distal foot. The clinical significance of such findings is thus uncertain and need to be further explored. Additionally, it should be noted that none of the markers of neutrophil activation were associated with worsening of ABI and 6MWD at follow-up. Taken together, these data seem to suggest a role for neutrophils in acute CV events, rather than in disease severity and its progression, in the setting of symptomatic chronic PAD.

Our study has several limitations worth noting. First, the small sample size and low number of outcomes significantly limit the interpretability of our findings. Second, our results cannot be extended to all subjects with PAD, as we did not enroll patients with asymptomatic disease (Leriche–Fontaine stage I) nor rest pain in legs (Leriche–Fontaine stage III), and only one patient displayed PAD stage IV. The measurement of the above markers could be particularly important in patients with asymptomatic disease, who represent the majority of PAD patients and whose CV risk may be similar to that of symptomatic patients [50]. Third, the great majority on MACE and/or MALE at 6-month follow-up consisted in disabling claudication or severe limb ischemia leading to an intervention, whereas myocardial infarctions were scarcely represented, and no patient developed stroke. Our results should thus be verified in larger studies with longer follow-ups, in order to ascertain the predictive value of the markers of neutrophil activation in the whole spectrum of adverse CV events. Another limitation is that the implementation of MPO–DNA measurements in clinical routine lacks standardization. Furthermore, the inclusion in regression analyses of diabetes, hypertension, and hypercholesterolemia as dichotomous covariates is a potential bias. However, we could not use them as continuous variables, as these might be strongly confounded by antidiabetics, antihypertensive, and lipid-lowering therapies. Lastly, our study was mostly descriptive and it is not possible to determine the pathomechanisms underlying neutrophil degranulation and NET generation in patients with PAD. Basic science studies are warranted to better understand these aspects.

Despite these limitations, our study has strengths. To the best of our knowledge, this is the first study assessing a large spectrum of markers of neutrophil activation, including a highly specific surrogate marker of NET generation, in patients with symptomatic chronic PAD compared with healthy controls. Moreover, the prospective nature of the study and the inclusion of a comprehensive vascular assessment of patients with PAD are additional strengths of our study and significantly increase the relevance of our findings.

## 5. Conclusions

Our study suggests enhanced neutrophil degranulation in patients with symptomatic chronic PAD compared with healthy subjects, even after adjustment for conventional CV risk factors. Conversely, NETs generation may be similar in the two groups, at least in stable conditions. In patients with symptomatic chronic PAD, biomarkers of neutrophil activation may not correlate with disease severity but rather be associated with an increased risk of acute CV events. These data suggest a role of neutrophils in PAD natural history that needs to be further elucidated.

## Figures and Tables

**Figure 1 biomedicines-11-00866-f001:**
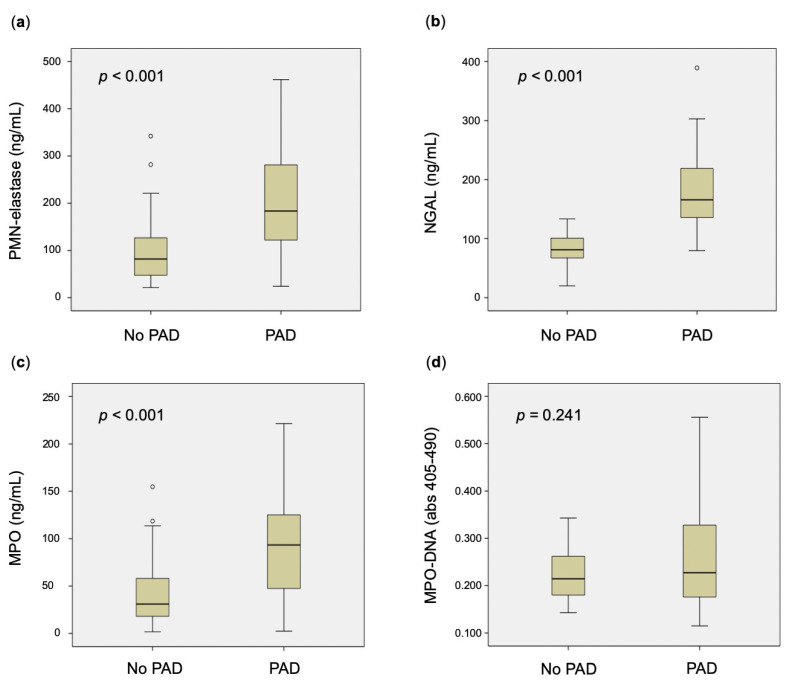
Markers of neutrophil activation, namely PMN-elastase (**a**), NGAL (**b**), MPO (**c**), and MPO–DNA (**d**) in subjects with and without symptomatic chronic PAD. Outliers are represented as small circles. Extreme outliers (95th percentile) may not be displayed in the figures. MPO: myeloperoxidase; NGAL: neutrophil gelatinase-associated lipocalin; PAD: peripheral artery disease; PMN: polymorphonuclear neutrophil.

**Figure 2 biomedicines-11-00866-f002:**
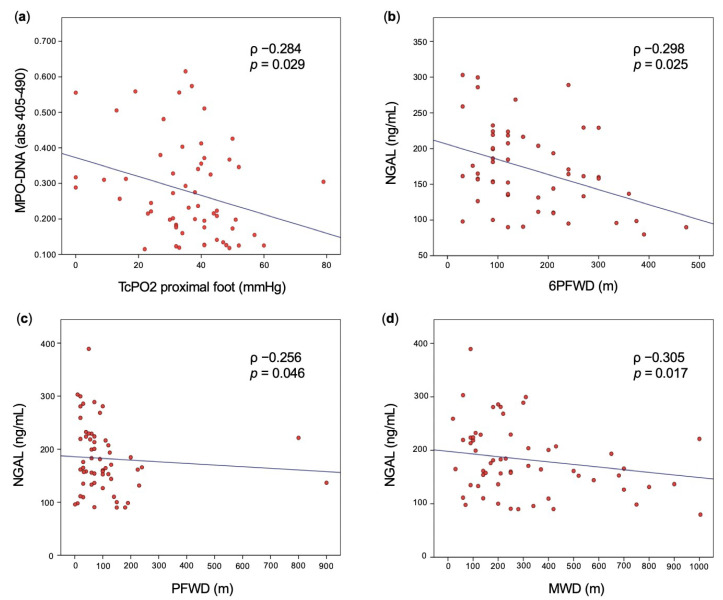
Scatter plot. Linear correlation analyses showing significant correlation between markers of neutrophil activation and vascular tests in patients with symptomatic chronic PAD: MPO–DNA and TcPO2 on proximal foot (**a**); NGAL and 6PFWD (**b**), PFWD (**c**), and MWD (**d**). Extreme outliers (95th percentile) may not be displayed in the figures. MWD: maximal walking distance; NGAL: neutrophil gelatinase-associated lipocalin; PFWD: pain-free walking distance; 6MWD: 6-min maximal walking distance; 6PFWD: 6-min pain-free walking distance.

**Figure 3 biomedicines-11-00866-f003:**
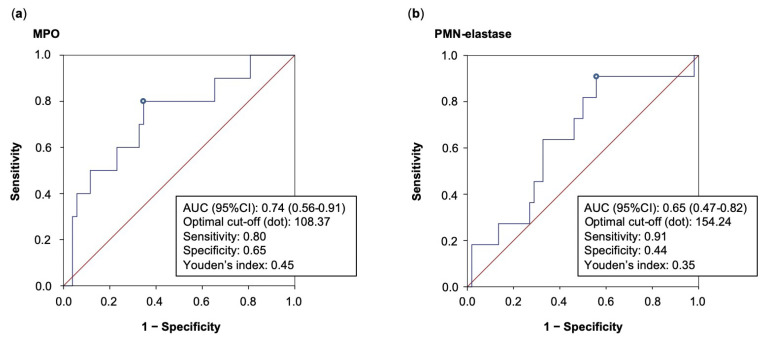
AUC and optimal cut-off values obtained from ROC curve analysis of biomarkers with best prognostic performance in terms of MACE and/or MALE, namely MPO (**a**) and PMN-elastase (**b**). AUC: area unde the curve; MPO: myeloperoxidase; PMN: polymorphonuclear neutrophil.

**Table 1 biomedicines-11-00866-t001:** Clinical characteristics and markers of neutrophil activation in subjects with and without symptomatic chronic PAD.

	All Subjects (=110)	PAD (=66)	No PAD (=44)	*p*
**Clinical characteristics**				
Sex (male) (%)	77 (70.0)	45 (68.2)	32 (72.7)	0.766
Age (years) (±SD)	59.1 (11.0)	64.2 (8.9)	51.3 (8.9)	<0.001
BMI (kg/m2) (±SD)	27.3 (5.0)	27.0 (5.2)	27.7 (4.7)	0.287
Waist (cm) (±SD)	99.1 (14.7)	100.0 (15.9)	97.9 (12.8)	0.918
**Comorbidities**				
Obesity (%)	24 (21.8)	14 (21.2)	10 (22.7)	1.000
Diabetes (%)	26 (23.9)	24 (36.4)	2 (4.7)	<0.001
Active smoking (%)	47 (42.7)	34 (51.5)	13 (29.5)	0.037
Hypertension (%)	64 (58.2)	46 (69.7)	18 (40.9)	0.003
Family history of CV events (%)	45 (42.5)	25 (40.3)	20 (45.5)	0.691
**Baseline treatment**				
Antihypertensive (%)	64 (58.2)	52 (78.8)	12 (27.3)	<0.001
Antiplatelet (%)	58 (52.7)	58 (87.9)	-	-
Statin (%)	60 (54.5)	57 (86.4)	3 (6.8)	<0.001
**Laboratory parameters**				
Creatinine (mg/dl) (±SD)	0.95 (0.32)	1.02 (0.38)	0.86 (0.17)	0.106
GFR (ml/min/1.73 m^2^) (±SD)	79.63 (22.21)	77.72 (23.15)	82.28 (20.83)	0.843
Total cholesterol (mg/dl) (±SD)	182.13 (40.39)	167.25 (31.76)	203 (42.11)	<0.001
LDL cholesterol (mg/dl) (±SD)	104.78 (37.68)	87.11 (28.73)	128.89 (35.19)	<0.001
HDL cholesterol (mg/dl) (±SD)	51.74 (28.28)	50.95 (18.37)	52.91 (38.86)	0.215
Triglycerides (mg/dl) (±SD)	156.20 (86.65)	152.91 (90.02)	160.91 (82.37)	0.229
**Markers of neutrophil activation**				
PMN-elastase (ng/mL) (±SD)	187.57 (163.91)	235.64 (182.03)	113.78 (92.75)	<0.001
NGAL (ng/mL) (±SD)	162.47 (150.19)	210.16 (175.46)	89.30 (36.08)	<0.001
MPO (ng/mL) (±SD)	83.16 (70.88)	105.15 (75.27)	49.12 (46.81)	<0.001
MPO–DNA (abs 405–490) (±SD)	0.257 (0.110)	0.275 (0.131)	0.231 (0.063)	0.241

BMI: body mass index; CV: cardiovascular disease; HDL: high-density lipoprotein; GFR: glomerular filtration rate; LDL: low-density lipoprotein; MPO: myeloperoxidase; NGAL: neutrophil gelatinase-associated lipocalin; PAD: peripheral artery disease; PMN: polymorphonuclear neutrophil; SD: standard deviation.

**Table 2 biomedicines-11-00866-t002:** Markers of neutrophil activation at baseline in patients with symptomatic chronic PAD with and without outcomes at 6 months.

	All Subjects (=63)	No Outcome (=52)	MACE and/or MALE (=11)	*p*	All Subjects (=49)	No Outcome (=35)	ABI Reduction ≥ 0.15 ^a^ (=14)	*p*	All Subjects (=40)	No Outcome (=30)	6MWD Reduction ≥ 20 m (=10)	*p*
PMN-elastase (ng/mL) (±SD)	240.39 (184.79)	227.27 (182.83)	302.37 (190.04)	0.128	204.88 (106.44)	206.67 (112.93)	201.19 (91.76)	0.765	207.21 (104.35)	215.20 (114.12)	183.25 (66.11)	0.508
NGAL (ng/mL) (±SD)	207.94 (178.06)	188.69 (97.55)	298.93 (370.14)	0.269	182.56 (99.35)	173.18 (81.74)	201.91 (136.76)	0.609	176.47 (83.71)	181.71 (89.55)	160.74 (64.46)	0.634
MPO (ng/mL) (±SD)	106.85 (76.14)	98.52 (74.83)	150.17 (71.26)	0.020	98.25 (74.69)	90.54 (61.48)	114.14 (97.18)	0.406	95.28 (62.70)	94.85 (58.66)	96.58 (77.10)	0.794
MPO–DNA (abs 405–490) (±SD)	0.275 (0.133)	0.256 (0.126)	0.362 (0.132)	0.014	0.250 (0.114)	0.262 (0.105)	0.227 (0.129)	0.209	0.246 (0.126)	0.252 (0.124)	0.230 (0.135)	0.528

^a^ Or TBI reduction ≥ 0.1. ABI: ankle-brachial index; MACE: major adverse cardiovascular event; MALE: major adverse limb event; MPO: myeloperoxidase; NGAL: neutrophil gelatinase-associated lipocalin; PMN: polymorphonuclear neutrophil; SD: standard deviation; TBI: toe-brachial index; 6MWD: 6-min maximal walking distance.

**Table 3 biomedicines-11-00866-t003:** Cox and logistic regression analyses of markers of neutrophil activation for outcomes at 6-month follow-up in patients with symptomatic chronic PAD.

	MACE and/or MALE						ABI reduction ≥ 0.15 or TBI reduction ≥ 0.1			6MWD reduction ≥ 20 m		
	Univariable Analyses			Multivariable Analyses ^a^			Univariable Analyses			Univariable Analyses		
	HR	95%CI	*p*	HR	95%CI	*p*	HR	95%CI	*p*	HR	95%CI	*p*
PMN-elastase (ng/mL) (±SD)	1.003	1.000–1.007	0.077	1.010	1.000–1.020	0.040	1.000	0.994–1.005	0.864	0.997	0.989–1.004	0.400
NGAL (ng/mL) (±SD)	1.001	0.999–1.003	0.358		-		1.003	0.997–1.009	0.354	0.996	0.987–1.007	0.996
MPO (ng/mL) (±SD)	1.027	1.004–1.051	0.019	1.027	1.004–1.051	0.019	1.004	0.996–1.012	0.311	1.000	0.989–1.012	0.939
MPO–DNA (abs 405–490) (±SD)	0.960 *	0.904–1.019	0.178		-		0.971 *	0.915–1.030	0.324	0.985 *	0.926–1.048	0.640

CI: confidence interval; HR: Hazard ratio; MPO: myeloperoxidase; NGAL: neutrophil gelatinase-associated lipocalin; OR: Odds ratio; PMN: polymorphonuclear neutrophil. ^a^ Including: age, hypertension, diabetes, and active smoking. * For each 0.01-unit increase.

## Data Availability

The data in this study are not publicly available but may be provided after reasonable request. Please contact Dr. Giacomo Buso (giacomo.buso@unil.ch) for further information.

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
