# Peer review of "Biomarkers of Neutrophil Activation in Patients with Symptomatic Chronic Peripheral Artery Disease Predict Worse Cardiovascular Outcome"

_biomedicines, 2023, doi:10.3390/biomedicines11030866_

Round 1

Reviewer 1 Report

Neutrophils have been implicated in atherosclerosis but their detailed role in PAD pathogenesis is not well documented. Buso et al. present in their manuscript a study exploring the prognostic value of circulating markers of neutrophil activation for MACE and MALE in patients with symptomatic PAD. They report that PMN elastase and MPO are predictive of 6-month MACE and/or MALE. Overall, the work is well done.

Concerns and suggestions

-In the manuscript there is no description of neutrophil activity assessment but rather the measurement of markers of neutrophil activation. In view of this, I feel the wording of “Enhanced neutrophil activity” in the title should be modified. 

-The authors discussed the work by Shahab et al. (Vascular. 2021;29:363-371) short of stressing the prognostic value of serum MPO levels for MACE and MALE in patients with PAD shown by the original authors.

-Is “2%” (line 93 on page 2) the estimated death probability by the GRACE Score?

-To determine the performance of MPO and PMN elastase in predicting the events (e.g., sensitivity and specificity), I suggest the authors conduct the receiver operating characteristic (ROC) curve analysis and obtain an area under the curve (AUC) value for each marker.

Author Response

Neutrophils have been implicated in atherosclerosis but their detailed role in PAD pathogenesis is not well documented. Buso et al. present in their manuscript a study exploring the prognostic value of circulating markers of neutrophil activation for MACE and MALE in patients with symptomatic PAD. They report that PMN elastase and MPO are predictive of 6-month MACE and/or MALE. Overall, the work is well done.

We are very thankful to the Reviewer for this kind remark.

Concerns and suggestions

  • In the manuscript there is no description of neutrophil activity assessment but rather the measurement of markers of neutrophil activation. In view of this, I feel the wording of “Enhanced neutrophil activity” in the title should be modified.

We fully agree with Reviewer's concern. Accordingly, we have modified the title as follows: “Biomarkers of neutrophil activation in patients with symptomatic chronic peripheral artery disease predict worse cardiovascular outcome”. We have also modified the Conclusions section as follows: “In patients with symptomatic chronic PAD, biomarkers of neutrophil activation may not correlate with disease severity but rather be associated with an increased risk of acute CV events” (page 12, lines 490-492).

  • The authors discussed the work by Shahab et al. (Vascular. 2021;29:363-371) short of stressing the prognostic value of serum MPO levels for MACE and MALE in patients with PAD shown by the original authors.

We thank the Reviewer for the suggestion. In order to further emphasize the results of Schahab et al. we have modified the corresponding text in the Discussion section as follows: “Intriguingly enough, in a recent prospective cohort study including patients with PAD undergoing digital subtraction angiography, baseline MPO levels were 3.68 times higher in patients with all-cause death and MACE, and 1.48 times higher in those with MALE than those without such outcomes at 24 months follow-up. Moreover, the authors found higher MPO levels in patients with multi-bed vascular disease compared with those with PAD alone, suggesting that this biomarker may reflect the extent of vascular damage in patients with atherosclerotic disease” (page 10, lines 372-379).

  • Is “2%” (line 93 on page 2) the estimated death probability by the GRACE Score?

That is correct. In order to avoid confusion, we have modified the test as follows: “[…] estimated mortality risk <2% according to the GRACE Risk Score” (page 3, lines 98-99).

  • To determine the performance of MPO and PMN elastase in predicting the events (e.g., sensitivity and specificity), I suggest the authors conduct the receiver operating characteristic (ROC) curve analysis and obtain an area under the curve (AUC) value for each marker.

We totally agree with the Reviewer on this aspect. We have therefore added this additional analysis to our work, as highlighted in the Abstract (page 1, lines 33-36), Materials and Methods (page 5, lines 203-207), Results (page 9, lines 340-344), and Discussion (page 11, lines 387-389). We have also added a figure concerning these results (Figure 3).

Reviewer 2 Report

The authors have presented a very interesting study. The results are new and highly significant. I have a few minor comments:

1. Page 2 line 49 - typos "leucocytes"

2. Please add the catalog numbers of the ELISA kits.

3. Page 6 line 242-245 - What was included in the multivariate analysis?

4. After the ninth page, the numbering is confused and the discussion begins on page 1. So, in the discussion, Page 2 line 49, he authors indicate that a correction has been made for conventional risk factors. However, as far as I understood, these conventional factors included only - age, hypertension, smoking and diabetes?

Author Response

The authors have presented a very interesting study. The results are new and highly significant.

We are very thankful to the Reviewer for this kind remark.

I have a few minor comments:

  1. Page 2 line 49 - typos "leucocytes"

We thank the Reviewer for the suggestion. We have corrected the typo accordingly.

  1. Please add the catalog numbers of the ELISA kits.

We have added the cat. numbers of the ELISA kits in the text, as requested.

  1. Page 6 line 242-245 - What was included in the multivariate analysis?

We agree with the Reviewer that this information should be specified in the text. We have therefore added the following sentence in the corresponding section: “Uni- and multivariable regression analyses, the latter including age, diabetes, hypertension, and active smoking as covariates, confirmed a significant association with […]” (page 6, lines 253-254).

  1. After the ninth page, the numbering is confused and the discussion begins on page 1. So, in the discussion, Page 2 line 49, he authors indicate that a correction has been made for conventional risk factors. However, as far as I understood, these conventional factors included only - age, hypertension, smoking and diabetes?

We thank the Reviewer for the remarks. First, we corrected the line numbering in the text. With regard to the second point, a correction was made only for conventional risk factors found to be significantly associated with PAD (significance level of p<0.10) on bivariate analysis, as per our statistical approach (page, lines). Such risk factors are indeed age, diabetes, hypertension, and active smoking. Other potential cardiovascular risk factors were not investigated and we agree that this should be better specified in the text. Accordingly, we have modified the sentence as follows: “Adjustment for conventional risk factors associated with PAD presence on bivariate analysis, namely age, diabetes, hypertension, and active smoking did not attenuate the association […]” (page 10, lines 382-384).

Reviewer 3 Report

This Manuscript by Buso G et al. is well written, nicely presented and interesting. 

Please add country in line 85.

Each section is perfectly organized and I do not have any further comments.

Author Response

This Manuscript by Buso G et al. is well written, nicely presented and interesting.

We are very thankful to the Reviewer for this kind remark.

Please add country in line 85.

We have added the country and the city of Cà Foncello University Hospital, as requested (page 2, lines 90-91).

Each section is perfectly organized and I do not have any further comments.

We truly appreciate the Reviewer’s kind remark.